# ROBUST MULTI-AGENT REINFORCEMENT LEARNING AGAINST ADVERSARIES ON OBSERVATION

## ABSTRACT

With the broad applications of deep learning, such as image classification, it is becoming increasingly essential to tackle the vulnerability of neural networks when facing adversarial attacks, which have been widely studied recently. In the cooperative multi-agent reinforcement learning field, which has also shown potential in real-life domains, little work focuses on the problem of adversarial attacks. However, adversarial attacks on observations that can undermine the coordination among agents are likely to occur in actual deployment. This paper proposes a training framework that progressively generates adversarial attacks on agents' observations to help agents learn a robust cooperative policy. One attacker makes decisions on a hybrid action space that it first chooses an agent to attack and then outputs the perturbation vector. The victim policy is then trained against the attackers. Experimental results show that our generated adversarial attacks are diverse enough to improve the agents' robustness against possible disturbances.

## 1 INTRODUCTION

The target of reinforcement learning (RL) is to learn the policies in complex environments to get long-term rewards. The technique of MARL is introduced to adapt RL algorithms to multi-agent systems. In recent years, Multi-Agent Reinforcement Learning (MARL) has attracted widespread attention (Du & Ding, 2021) and has been applied in numerous domains, including sensor networks (Zhang & Lesser, 2011), autonomous vehicle teams (Zhou et al., 2020), and traffic signal control (Du et al., 2021). However, neural networks are proven to be vulnerable to adversarial perturbations (Huang et al., 2017), and some small perturbations may cause the deep RL policy to fail. Therefore, it is of great significance to train a robust policy to help deploy current RL algorithms to real-life applications.

In single-agent reinforcement learning, some research studies enhance policy robustness by using adversarial learning and achieve good results. Pinto et al. (2017) propose a method that jointly trains a pair of agents, including a protagonist and an adversary, and the protagonist learns to fulfill the original task goals while being robust to the disruptions generated by its adversary. Pattanaik et al. (2018) show that deep RL can be fooled easily and train an RL agent under naive attacks to improve its robustness. Zhang et al. (2021) propose a framework of alternating training with learned adversaries, which trains an adversary online with the agent using a policy gradient following the optimal adversarial attack framework. However, such studies are rare in cooperative MARL, and current works mainly focus on the setting where teammates may betray or agents' actions may be maliciously modified (Li et al., 2019; Phan et al., 2021; 2020; Hu & Zhang, 2022). However, in real-life applications of cooperative MARL, the most vulnerable parts of the agents are the sensors that can be disturbed by noise or jamming attacks. Agents are closely related to each other when cooperating to accomplish tasks, and even a small perturbation on one agent's observation from the sensors can make it deflect from coordination and cause the whole multi-agent system to fail. Therefore, how to design an algorithm that can obtain a policy that is robust on observations in cooperative MARL is noteworthy.

This paper proposes a robust MARL training framework for observation perturbations, RObust Multi-agent reinforcement learning against Adversaries on Observation (ROMAO). Our contributions can be summarized as follows:

- We propose a hybrid action attacker that attacks based on the global state. Its output consists of a discrete action and a continuous action. The discrete action refers to which agent to disturb, and the continuous action refers to the disturbance added to the observation of that agent.

- We propose a robust training framework that generates attackers gradually and helps the victim team learn a robust cooperative policy that is resistant to perturbations on observations.

- We additionally propose an optional defense module that can further enhance agents' ability to defend against attacks on observations.

- Experimental results show that the proposed attacker can pinpoint the weakness of the cooperative policies, and our robust training framework along with the defense module can effectively improve the robustness of the policies against possible attacks.

## 2 RELATED WORK

### 2.1 COOPERATIVE MULTI-AGENT REINFORCEMENT LEARNING

Cooperative MARL has made prominent progress these years. Research on it aims to help agents learn policies to coordinate and complete cooperative tasks. Many methods have emerged under the CTDE paradigm, most of which can be roughly divided into policy-based and value-based methods. MADDPG (Lowe et al., 2017), COMA (Foerster et al., 2018), and MAAC (Iqbal & Sha, 2019) are typical policy gradient-based methods that explore the optimization of multi-agent policy gradient methods, while MADDPG can also be employed in competitive scenarios. Another category of cooperative MARL approaches, value-based methods, mainly focus on factorizing the value function. VDN (Sunehag et al., 2018) aims to decompose the team value function into agent-wise value functions by a simple additive factorization. Following the Individual-Global-Max (IGM) principle (Son et al., 2019), QMIX (Rashid et al., 2018) improves the way of value function decomposition by learning a mixing network, which approximates a monotonic function value decomposition.

### 2.2 ADVERSARIAL ATTACK

The adversarial attack has been explored in many areas. In image classification, the adversarial attack means generating adversarial examples. The adversarial example is a deceptive input to a model that is purposely designed to cause a model to make a mistake in its predictions but makes no difference to humans. Goodfellow et al. (2015) propose a simple and fast gradient-based method that is used to generate adversarial examples to make the model classify incorrectly while minimizing the amount of perturbation added to the pixels of the image. Loison et al. (2020) use feature selection to minimize the number of features modified while causing the wrong classification, and flat perturbations are added to features iteratively according to saliency value by decreasing order.

### 2.3 ADVERSARIAL ROBUSTNESS OF RL AGENTS

Based on the effectiveness of adversarial attacks on images, Huang et al. (2017) propose a method to inject adversarial perturbation into the input to confuse the RL policy. Some researchers (Gleave et al., 2019; Zhao et al., 2020) focus on black box attacks in RL, which are more challenging because of the lack of information about the parameters of the target model. Adversarial training is empirically shown to improve agents' robustness to make the policies experience possible adversarial attacks. Pinto et al. (2017) propose a method to train an agent in the presence of disturbance and obtain more robust policies. Zhang et al. (2021) propose a method that involves the concurrent training of an attacker and the victim agent using policy gradient following the optimal adversarial attack framework. Sun et al. (2021) decouple the problem of finding state perturbations into finding the best policy perturbation directions and crafting correspondent state perturbations.

### 2.4 ADVERSARIAL ATTACKS IN COOPERATIVE MARL

There could be various types of adversarial attacks in cooperative multi-agent systems. Some researchers focus on the setting where some teammates may betray and minimize their shared re-

turn (Phan et al., 2021; 2020; Li et al., 2019). Meanwhile, some researchers prefer the setting where components of a Markov Decision Process (MDP), such as states, actions, or observations, are perturbed. Hu & Zhang (2022) propose a sparse adversarial attack on actions of cooperative multi-agent systems and can make the victim team perform poorly when only a few agents are attacked at a few timesteps. Zhou & Liu (2022) propose a robust training framework for the state-of-the-art reinforcement learning method MFAC (Yang et al., 2018) when the state is perturbed. Lin et al. (2020) propose a method to attack one agent's observation in a team. It is achieved by an indirect way that the attacker first tries to find a wrong action it should encourage the victim agent to take. Then, the attacker uses adversarial examples to mislead the victim into choosing the action. This work is most relevant to our work because it considers the indirect attacks on observations, and the attacker only chooses one agent to attack. While in our work, we consider the setting that every agent is at risk of attack, and we cannot access the parameters of the Q functions of the victims. We focus on black box attacks on observations and how to defend against them, which is more reasonable and realistic.

## 3 PROBLEM FORMULATION

This paper considers a fully cooperative multi-agent task where agents only have access to partial observations for the victim side. When the attacker's policy is fixed, the model is defined as $\mathcal{M} = \langle \mathcal{N}, \mathcal{S}, \{A^i\}_{i=1}^n, P, \{O^i\}_{i=1}^n, \Omega, R, \gamma \rangle$, where $\mathcal{N} = \{1, \ldots, n\}$ is the set of agents, $s \in \mathcal{S}$ is the true global state from which agent can get local observation $o^i \in O^i$. In order to relieve the partial observability problem, we add an RNN module, GRU (Cho et al., 2014), called agent trajectory encoder, to encode the history $(o_1^i, a_1^i, \ldots, o_{t-1}^i, a_{t-1}^i, o_t^i)$ into $\tau^i$, with $a_t^i \in A^i, o_t^i \in O^i$ stand for the action and observation of agent $i$ at time $t$. At each timestep, each agent selects an action $a_t^i \in A^i$, forming a joint action $\boldsymbol{a} \in A = \prod_{i=1}^n A^i$, leading to the next state $s_{t+1} \sim P(\cdot|s_t, \boldsymbol{a})$ and getting a shared reward $R(s_t, \boldsymbol{a})$. The formal objective for the multi-agent policy (the victim side) is to find a joint policy $\boldsymbol{\pi}(\boldsymbol{\tau}, \boldsymbol{a})$ to maximize the global value function:

$$Q_{\text{tot}}^{\boldsymbol{\pi}}(\boldsymbol{\tau}, \boldsymbol{a}) = \mathbb{E}_{\boldsymbol{\tau}, \boldsymbol{a}} \left[ \sum_{t=0}^{\infty} \gamma^t R(s_t, \boldsymbol{a}_t) \mid s_0 = s, \boldsymbol{a}_0 = \boldsymbol{a}, \boldsymbol{\pi} \right], \quad (1)$$

where $\gamma$ indicates the discount factor. Perturbations are integrated into $\{O^i\}_{i=1}^n$ and the agents cannot access the oracle information about whether they are under attack. Meanwhile, when the victim team's policy is fixed, the Markov Decision Process for the attacker is defined as $\hat{\mathcal{M}} = \langle \mathcal{S}, \hat{\mathcal{A}}, \hat{P}, \hat{R}, \gamma \rangle$. The attacker shares the same state space with the victim team and $\hat{\mathcal{A}} : \mathcal{N} \times \{O^i\}_{i=1}^n$, which means that the attacker first chooses one agent and then generates a perturbation with the same dimension as the agent's observation. The transition function is defined as $\hat{P} : \mathcal{S} \times \hat{\mathcal{A}} \times \{A^i\}_{i=1}^n \to \mathcal{S}$. Since $\boldsymbol{\pi}$ is fixed when training the attackers, $\boldsymbol{a} \in A$ is only determined by the state $s \in \mathcal{S}$. The reward function is $\hat{R} = -R$ and the objective of the attacker's policy $\hat{\pi}$ is to reduce the return of the victim team which is equivalent to maximizing the following objective ($\hat{a}_t \in \hat{\mathcal{A}}$):

$$J(\hat{\pi}) = \mathbb{E}_{\hat{\pi}} \left[ - \sum_{t=0}^{\infty} \gamma^t R(s_t, \boldsymbol{a}_t) \mid s_{t+1} \sim \hat{P}(\cdot|s_t, \hat{a}_t, \boldsymbol{a}_t), \hat{a}_t \sim \hat{\pi}(s_t, \cdot) \right]. \quad (2)$$

Attacks on multi-agent systems may be diverse. In some scenarios, random noise can be seen as a potential implicit attacker. While in others, the attacker can add perturbations to a specific observation dimension of a specific agent, which is small enough and will not make a difference to people but can seriously degrade the performance of multi-agent systems. We mainly consider the following kinds of possible attacks on multi-agent in our work:

- Attacking a random agent with random noise: The observation of each agent in a multi-agent system may contain random noise, which may be physically caused (e.g., sensor errors). We assume that the noise in each observation dimension of each agent is independently and identically distributed.

- Attacking a specific agent with random noise: In some scenarios, only one specific agent will contain random noise. We also assume the noise is independently and identically distributed in this attack setting.

- Attacking a specific observation dimension of a specific agent: In some scenarios, a specific observation dimension of a specific agent may be under attack (e.g., in soccer, a player injures one eye resulting in inaccurate observations of himself).

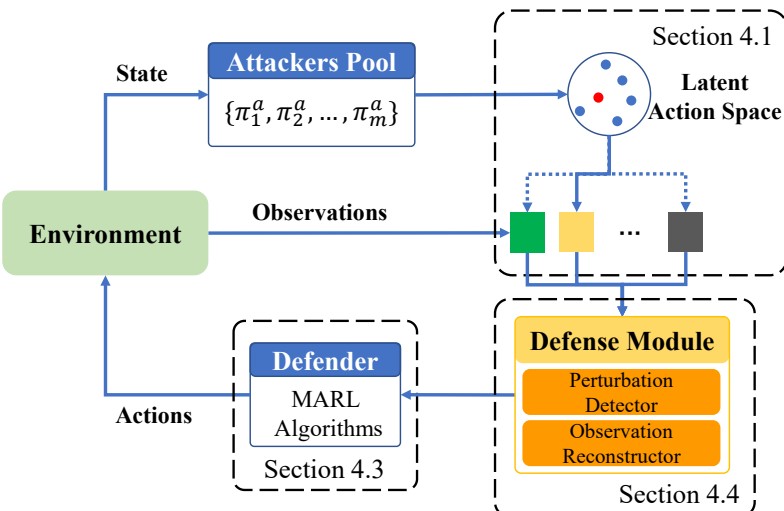

Figure 1: Structure of our robust training framework RAMAO. The attackers' pool stores history attacker models $\hat{\pi}_j, j = 1, 2, ..., m$, where $m$ is a hyperparameter we set about the maximum number of models the pool can store.

To be more consistent with the real-life problem setting, we assume that the $\ell_1$ norm of all attacks is less than or equal to a constant $C$. In general, the performance of a policy falls off a cliff when the policy is exposed to the above attacks. We want to obtain a robust policy that can prevent the performance from dropping too much under attacks with modes including but not limited to the above three.

## 4 METHOD

The overall structure of our proposed framework is shown in Figure 1, and we train the attacker and the defender (the victim team) alternatively. When training the attackers, we fix the defender and maintain the attackers' pool to store all the history attackers have trained so far. Every time the attacker starts a turn for training, it stores its last attack model $\hat{\pi}_j$ in the pool and resets its exploration and replay buffer to eliminate the bias introduced by previous attacks. During interactions, the HyAR attacker receives the global state, chooses an action in a latent space, and decodes the latent action to perturb the observation of a specific agent (Details are listed in Section 4.1). When training the defender, as stated in Section 4.3, we randomly sample an attacker from the attackers' pool and train the defender against it with only access to the perturbed observations. The defense module is optional because it needs communication and will be introduced in detail in Section 4.4.

### 4.1 HYBRID ACTION SPACE

In our method, the attacker first selects an agent to attack, then gives the offset for each dimension of the observation, which can be considered a hybrid action space problem because selecting an agent is a discrete action and giving the offset for each dimension of the observation is a continuous action. We will consider a Parameterized Action MDP (PAMDP) $\langle \mathcal{S}, \mathcal{H}, P, R, \gamma, T \rangle$ (Masson & Konidaris, 2015), which is an extension of MDP to consider the discrete-continuous hybrid action space $\mathcal{H} = \{(k, x_k) \mid x_k \in \mathcal{X}_k, \forall k \in \mathcal{K}\}$, where $\mathcal{K} = \{1, \cdots, K\}$ is the set of the discrete action and $\mathcal{X}_k$ is the continuous parameter set for each $k \in \mathcal{K}$ (Li et al., 2021).

Typical policy representations like Gaussian distribution and Multinomial distribution also cannot model the heterogeneous components of the hybrid action space. A naive approach to solving the hybrid action problem is to convert the hybrid action space into a discrete action space by discretizing the continuous action space. However, this ignores the implicit connection between continuous and discrete actions. Meanwhile, it reduces scalability and brings approximation difficulties.

Li et al. (2021) propose a novel method named HyAR for solving hybrid action space. HyAR constructs a unified and decodable latent representation space ($\mathbb{R}^{d_1+d_2}$, $d_1$ is the latent space dimension for discrete action $k$ and $d_2$ is for continuous action $x_k$) for original hybrid actions, and the agent learns a latent policy. Then, with the decoder, we can map the selected latent action back to the original hybrid action space to interact with the environment. In more detail, HyAR establishes an embedding table $E_\zeta \in \mathbb{R}^{K \times d_1}$ to represent the $K$ discrete actions. Each row $e_{\zeta,k} = E_\zeta(k)$ is a $d_1$-dimensional embedding vector for the discrete action $k$. Then, with state $s$, HyAR uses an encoder $q_\phi(z \mid x_k, s, e_{\zeta,k})$ parameterized by $\phi$ to map $x_k$ to the latent vector $z_x \in \mathbb{R}^{d_2}$. RL algorithms for continuous actions like DDPG (Lillicrap et al., 2016) or TD3 (Fujimoto et al., 2018) can then be applied to train on this latent action space. For inference, each time HyAR policy receives a state $s$, it will output latent actions $e, z_x$, then use the decoder and embedding table to reconstruct the latent actions back to the origin hybrid actions $k, x_k$.

HyAR takes full advantage of environmental dynamics and considers the connection between discrete and continuous actions. In our work, we employ HyAR to help the attacker learn the hybrid action to inflict perturbations on agents' observations. Practically, we employ TD3 as the policy to decide on the latent space. Based on state $s_t$ at time $t$, our attacker policy first outputs a discrete action $k_t$ to indicate the agent's ID that we will attack. Then the attacker policy outputs a continuous action $x_{k_t}$ which has the same dimension as $o_{k_t}$. The victim agent does not have oracle information about whether it is under attack and would make decisions based on the perturbed observation $o_{k_t} + x_{k_t}$.

## 4.2 Attacker Optimization Method

The goal of the adversarial attacker is to minimize the victim team's expected return by perturbing the observations of a certain agent in a multi-agent system. We do not introduce any other intuitive rewards that are possible to influence the possible attack modes. During the alternative training between the attacker and the victim team, we reset the exploration of the attacker to reduce the bias from the last round of attack learning. While for the victim team, we save the attacker's model every round, and the victim team can train their cooperation policy against all historical attackers, which would further enhance the robustness of our victim policy.

## 4.3 Robust Training

As formulated above, the attacker can inflict perturbations of amount $C$ in terms of the $\ell_1$ norm on one agent's observation. However, we do not assume what kind of attacker pattern the attacker uses. In practice, an attacker may apply a random noise to attack the observation, while others can also concentrate on specific dimensions of the observation based on the current situation. The attack may even have a long-term effect that we should not only focus on one timestep when the agent's observation is perturbed. To discover the weakness of the victim team's policy and make agents get used to different kinds of perturbations, we employ the above attackers and alternate training between the attackers and the victim team. The advantage of alternate training is that when the defender has a high win rate, it forces the attacker to find some tricky attacks, and the defender can learn to deal with these tricky attacks, thus greatly increasing the ability to defend against both general attacks or certain extreme attacks. During the training of the victim team, we sample an attacker for an episode from a history pool of all attackers trained so far so that the victim team can defend against not only the current version of the attacker but also all the history attackers to improve its general robustness. Besides, our training framework is agnostic to specific MARL methods and can be applied to improve any MARL method's robustness.

## 4.4 Defense Module

Apart from the adversarial training, we propose a multi-agent system defense module to defend against the above perturbations. In multi-agent scenarios, agents interact in the same environment, so agents close to each other may share part of their observations. Inspired by this fact, we came up with a defense module that an agent can reconstruct its observations based on the observations of its teammates in its field of view. However, an agent cannot access the information about whether it is perturbed or not, so we need to train a perturbation detector to judge whether an agent is under

attack. The mentioned training can be achieved during the centralized training phase since both the original and the perturbed observations are available.

## 5 EXPERIMENTS

In this section, we first display the scenarios involved in our experiments in Section 5.1, including StarCraft II unit micromanagement benchmark (SMAC) (Samvelyan et al., 2019) and predator-prey (PP) (Boehmer et al., 2020). We show the main results and the effectiveness of our robust training framework in Section 5.2. Then we present part of our training curve on QMIX and employ Principal Component Analysis (PCA) (Pearson, 1901) to illustrate the diversity of our generated attackers in Section 5.3. Furthermore, we demonstrate the effects of our defense module in Section 5.4 and the generality of our trained robust cooperative policy.

We choose QMIX (Rashid et al., 2018), which is a popular baseline in cooperative MARL methods and approximates a monotonic function value decomposition for joint value function, as our victim team policy, while our robust training framework ROMAO is agnostic to specific MARL algorithms or attacker policies. ROMAO is a general framework that can be employed to improve the robustness of any given MARL method against perturbations on observations. More details about our experiments would be listed in the appendix.

### 5.1 ENVIRONMENTS

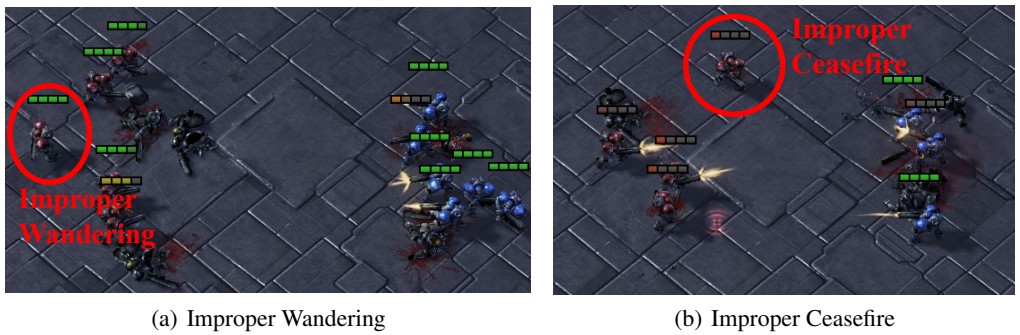

(a) Improper Wandering                    (b) Improper Ceasefire

Figure 2: Agents' improper behavior patterns caused by observation perturbations.

We evaluate our robust training framework in environments including StarCraft II unit micromanagement benchmark (SMAC) and predator-prey (PP). More detailed descriptions of these environments can be found in the appendix.

#### 5.1.1 STARCRAFT II MICROMANAGEMENT BENCHMARK (SMAC)

SMAC (Figure 5.1) is a popularly used combat scenario of StarCraft II unit micromanagement tasks, where we train the ally units to beat enemy units controlled by the built-in AI with an unknown strategy. At each timestep, agents can move or attack any enemies and get a global reward equal to the total damage done to enemy units. Figure 5.1 shows the map of 5m_vs_6m in SMAC which indicates that we control 5 marines to defeat 6 enemy marines. It is a hard map in SMAC that the marines we control must work closely to win the battle but if one of the teammates is perturbed and does not cooperate normally as shown in Figure 2(a)&2(b), the other agents would be short of fire and in great danger.

#### 5.1.2 PREDATOR PREY (PP)

PP (Boehmer et al., 2020) is a partially observable grid-world task where $m$ predators (agents) are trained to capture $n$ moving preys. Agents get rewards only when there are no empty grids around the prey and at least two predators adjacent to it execute "catch" action concurrently. Miscatching would lead to a punishment of $-2$, this punishment makes it a challenging benchmark for current MARL methods.

| | Maps Methods | 2s3z | 3m | 3s_vs_5z | 5m_vs_6m |
|---|---|---|---|---|---|
| Attack Mode 1 | Vanilla QMIX | 95.3±3.4 | 91.1±1.5 | 81.2±4.4 | 58.3±3.7 |
| | Random QMIX | 98.9±0.9 | **97.2±0.9** | 97.6±0.9 | 65.4±1.6 |
| | One-agent QMIX | 81.4±2.3 | 86.0±1.8 | 77.9±3.0 | 59.2±6.8 |
| | ROMAO | **99.0±0.7** | 93.9±1.7 | **98.1±0.8** | **68.8±2.3** |
| Attack Mode 2 | Vanilla QMIX | 92.7±2.9 | 91.7±3.2 | 77.6±1.5 | 57.3±3.2 |
| | Random QMIX | **99.0±0.7** | **97.9±1.0** | **98.1±1.0** | **67.8±4.0** |
| | One-agent QMIX | 82.9±2.7 | 86.1±3.5 | 82.2±2.7 | 60.1±2.7 |
| | ROMAO | 96.3±1.0 | 94.9±2.1 | 97.8±0.7 | 66.0±2.3 |
| Attack Mode 3 | Vanilla QMIX | 72.6±5.9 | 5.73±2.0 | 45.8±4.1 | 13.5±0.7 |
| | Random QMIX | 83.1±2.3 | 90.8±2.0 | 56.0±3.5 | 3.1±2.4 |
| | One-agent QMIX | 69.3±4.2 | 92.9±1.8 | 54.6±4.8 | 13.6±2.3 |
| | ROMAO | **94.6±2.8** | **94.4±1.9** | **75.3±2.6** | **23.3±3.1** |

Table 1: Average test win rates of different methods under various attack modes with limited perturbation range of amount 5 in terms of the $\ell_1$ norm. We consider three different and general attack modes. Vanilla QMIX means the QMIX policy trained without perturbations. Random QMIX means the policy trained under random perturbations similar to Attack Mode 2. One-agent QMIX means the policy trained with only agent 0 is under attack and the attack is trained with TD3. ROMAO means the QMIX policy enhanced with our robust training framework.

## 5.2 RESULTS AND ANALYSIS

Table 1 shows the main results of our experiments about the average test win rates and correspondent standard deviations of different methods under various attack modes with a limited perturbation range of amount 5 in terms of the $\ell_1$ norm. We use perturbation range 5 for testing to evaluate the policies' generality trained under perturbations of range 10. Attack mode 1 denotes the attack that at every timestep, the attacker randomly chooses an agent and generates random perturbations. Attack mode 2 denotes the attack that at the start of an episode, the attacker randomly chooses an agent and generates random perturbations on that agent's observation throughout the entire episode. Attack mode 3 denotes the attack that, at the start of an episode, the attacker randomly chooses an agent and generates perturbations on only one random dimension of the agent's observation, which means an offset of 5 is added to a random dimension of the observation. Vanilla QMIX denotes the QMIX policies trained without perturbations. Random QMIX denotes the QMIX policies trained under the attack that randomly chooses an agent at each timestep and inflict random perturbations of amount $C$ in terms of the $\ell_1$ norm on the agent's observation. One-agent QMIX denotes the QMIX policies trained under the attack that always chooses agent 0 and inflict random perturbations of amount $C$. In practice, we set $C = 10$. All the means and standard deviations are the results of five random seeds for each method.

As shown in Table 1, after training with ROMAO, the ROMAO-trained QMIX policy is robust enough to defend against possible attacks and has a good and stable performance compared with QMIX policies that are trained in other ways. Generally, the vanilla QMIX performs worst since it has not seen any attack patterns before the evaluation, but it is still robust to some extent. It may be attributed to the fact that in a multi-agent system, even if one agent shuts down, other agents can still complete some simple tasks that are not demanding coordination. Training the MARL algorithms under random perturbations can also improve the policies' robustness, but they cannot deal with all possible situations. Random QMIX is trained under random perturbations like Attack Mode 2 so that it performs best on it, but it cannot generalize to other attacks and there is a significant drop in performance in Attack Mode 3, which is an extreme type of attack, while RAMAO-trained policy performs well on all these attacks. Attacking only one agent is also insufficient to generate all possible situations under attack because the attacking target can be switched among different agents. Due to the alternate training of ROMAO, it can cover more types of attacks and has better performance for extreme attacks like Attack Mode 3. The results demonstrate the necessity to employ a hybrid action space to generate diverse and influential attackers and the effectiveness of our proposed robust training framework.

### 5.2.1 RESULTS ON PP

We also carry out some experiments on Predator-Prey (PP). RO-MAO denotes the QMIX policy after the robust training in our proposed framework where attackers generate perturbations of amount $C$ in terms of the $\ell_1$ norm. RanPert denotes the QMIX policy trained under random perturbations of amount $C$. QMIX denotes the QMIX policy that is trained without any perturbations. The policies are evaluated under perturbations of a different range. In practice, we set $C = 10$ for training and $C = 5$ for this evaluation. As illustrated in Figure 3, ROMAO policies perform best, showing our proposed framework's effectiveness. During the training phase of our robust training framework, attackers generate diverse attack modes that can uncover different weaknesses of the current policy, while random perturbation cannot cover all the possible situations.

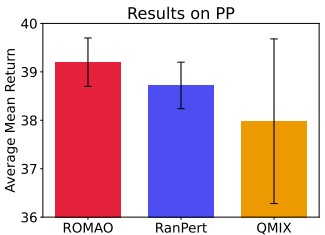

Figure 3: Results on Predator-Prey.

### 5.3 TRAINING PROCEDURE VISUALIZATION

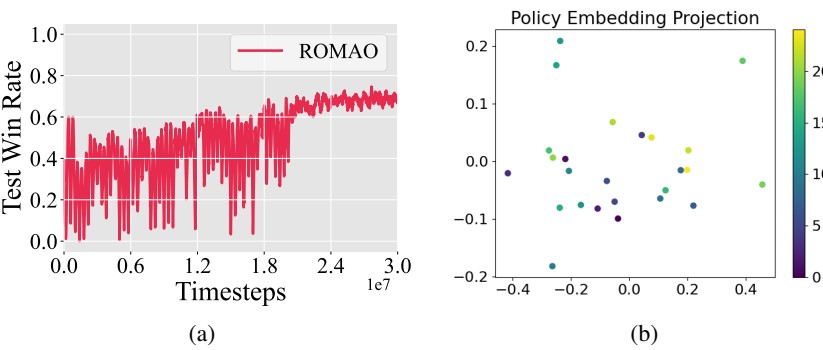

Figure 4: (a) The alternating training curve of the victim policy's performance. (b) The distribution of the representations of the attackers' policies. The darker the color of the dot, the earlier the attacker policy is generated.

Figure 4(a) shows the alternating training curve for QMIX on the map of 5m_vs_6m in SMAC under our framework ROMAO. We first train QMIX on 5m_vs_6m for 2M timesteps and get an average win rate of around 80%, and then load the trained model into the ROMAO framework to start alternating training. The curve shows the win rate of the current QMIX policy under the attack of the current attackers when testing. The attacker is trained in the first turn so that the win rate gets reduced first and fluctuates during the alternating training. Finally, the robust policy can maintain a win rate of over 60% however the attacker attacks.

We also illustrate the diversity of attacker policies in Figure 4(b). As stated in Section 4, we maintain an attackers' pool to store the previously trained attackers. As the defender will learn to adapt to the attackers' perturbations, the current attack mode may expire, and the attacker must seek a new attack mode to perturb the victim team's observations. In practice, we reset the attacker's exploration and replay buffer to eliminate the bias introduced by previous attacks. In order to achieve high rewards, the attackers should spontaneously find diverse attack modes to perturb the observations of the constantly updated victim agents.

Inspired by the recent success of transformers in deep learning (Vaswani et al., 2017), we utilize a transformer to encode a perturbed trajectory to represent an attack policy. We first sample $n$ trajectories of states $\{s_t\}_{t=0}^{T}$ ($T$ is the episode length) when the trained QMIX interacts with the environment under random perturbations. With the instinctive ability of a transformer to deal with variable-length input sequences, we concurrently train a transformer encoder that inputs a trajectory $\{s_t\}_{t=0}^{T-1}$ and outputs an embedding $h$ and a Multilayer Perceptron (MLP) decoder to reconstruct $s_T$ from $h$. After training, we fix the transformer encoder and get an attacker's policy embedding by first sampling 16 trajectories of vanilla QMIX under the attack of this attacker and then calculating the

mean of the outcomes after feeding the trajectories into the encoder. Finally, we use PCA to project the policies' embeddings on a 2-dimensional plane. As illustrated in Figure 4(b), we show the embeddings of the first 25 attacker policies. The darker the color, the earlier the policy is generated. We can see that the attackers can generate diverse attack policies spontaneously without explicit encouragement like intrinsic rewards to continuously lower the victim team's win rate.

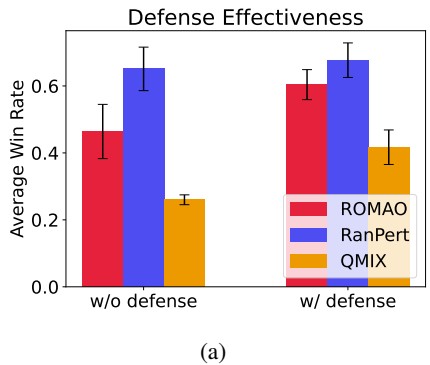
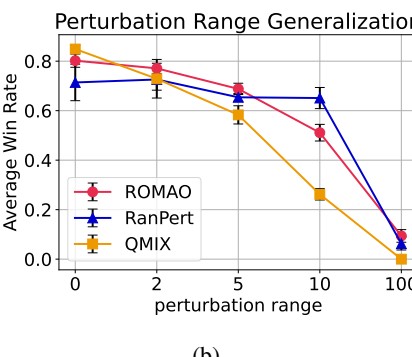

(a)                                                    (b)

Figure 5: (a) Policies' performance without and with the defense module. (b) Policies' performance under different perturbation ranges. ROMAO denotes the QMIX policy enhanced by our proposed framework. RanPert denotes the policy trained under random perturbations of amount $C$ in terms of the $\ell_1$ norm. QMIX denotes the QMIX policy that is trained with no perturbations.

## 5.4   DEFENSE MODULE

Figure 5(a) shows the effectiveness of our proposed defense module on QMIX in the map of 5m_vs_6m in Section 4.4. The win rate in the figure is tested under the random perturbation of amount 10 in terms of the $\ell_1$ norm, and the left three bars show the win rates of the trained policies without the defense module, and the right three show the improved performance with the defense module. RanPert achieves the best performance among the three policies because, in practice, we train RanPert under random perturbations of amount $C = 10$. It means that the training and testing environments are the same for RanPert so that it can achieve excellent performance. However, in reality, we cannot know in priority what kinds of perturbations in store our policies would be facing. The performances of all these policies are improved with the defense module without any further learning, demonstrating the effectiveness of our proposed defense module.

## 5.5   POLICY GENERALITY

As illustrated in Figure 5(b), we show the performances of different policies under different ranges of random perturbations in the map of 5m_vs_6m. ROMAO, RanPert, and QMIX are trained in the same ways as in Section 5.4. We can see that QMIX and Ranpert perform best as the testing settings are the same as their training settings (perturbations of amount 0 and 10). Under other circumstances, ROMAO, the QMIX policy after the training of our robust framework, achieves the best performance even under the perturbations of 100.

## 6   CONCLUSION

This paper proposes a new robust MARL training framework, RAMAO, for observation perturbations, which any MARL algorithm can employ to improve its robustness against possible attacks on agents' observations. The experimental results show that the attacker can generate diverse attack patterns to excavate the weaknesses of the victim MARL policy. After training, the MARL policy can defend against various kinds of perturbations, and the defense can be further enhanced with our proposed defense module. Future work can concentrate on how to deal with concurrent attacks on multiple agents efficiently as the combination blast cannot be avoided to attack multiple agents in ROMAO when the number of agents increases.

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

# A APPENDIX

## A.1 MORE RELATED WORK

### A.1.1 COOPERATIVE MULTI-AGENT REINFORCEMENT LEARNING

We introduce more about the value function factorization methods like QMIX in MARL here. VDN (Sunehag et al., 2018) and QMIX (Rashid et al., 2018) are two value-based factorization methods in cooperative MARL and follow the Individual-Global-Max (IGM) (Son et al., 2019) principle, which asserts the consistency between joint and local greedy action selections by the joint value function $Q_{\text{tot}}(\boldsymbol{\tau}, \boldsymbol{a})$ and individual value functions $\left[Q^i(\tau^i, a^i)\right]_{i=1}^n$:

$$\forall \boldsymbol{\tau} \in \mathcal{T}, \underset{\boldsymbol{a} \in \mathcal{A}}{\arg \max} Q_{\text{tot}}(\boldsymbol{\tau}, \boldsymbol{a}) =$$
$$\left(\underset{a^1 \in \mathcal{A}}{\arg \max} Q_1\left(\tau^1, a^1\right), \dots, \underset{a^n \in \mathcal{A}}{\arg \max} Q_n\left(\tau^n, a^n\right)\right). \tag{3}$$

VDN (Sunehag et al., 2018) factorizes the global value function $Q_{\text{tot}}^{\text{VDN}}(\boldsymbol{\tau}, \boldsymbol{a})$ as the sum of all the agents' local value funtions $\left[Q^i(\tau^i, a^i)\right]_{i=1}^n$:

$$Q_{\text{tot}}^{\text{VDN}}(\boldsymbol{\tau}, \boldsymbol{a}) = \sum_{i=1}^n Q_i\left(\tau^i, a^i\right). \tag{4}$$

QMIX (Rashid et al., 2018) extends VDN by factorizing the global value function $Q_{\text{tot}}^{\text{QMIX}}(\boldsymbol{\tau}, \boldsymbol{a})$ as a monotonic combination of the agents' local value funtions $\left[Q^i(\tau^i, a^i)\right]_{i=1}^n$:

$$\forall i \in \mathcal{N}, \frac{\partial Q_{\text{tot}}^{\text{QMIX}}(\boldsymbol{\tau}, \boldsymbol{a})}{\partial Q_i\left(\tau^i, a^i\right)} > 0. \tag{5}$$

We mainly test our framework ROMAO on QMIX for its proven performance in various papers. QMIX uses a hyper-net conditioned on the global state to generate the weights and biases of the local Q-values and uses the absolute value operation to keep the weights positive to guarantee monotonicity.

### A.1.2 ADVERSARIAL ATTACKS IN COOPERATIVE MARL

Some researchers focus on the setting where some teammates may betray and minimize their shared return. Phan et al. (2021) and Phan et al. (2020) propose to train competing teams of protagonist and antagonist agents of varying sizes to improve resilience against arbitrary agent changes. Li et al. (2019) extend MADDPG with a minimax objective to make the learned policy robust and behave well even with strategies not seen during training. Agents update policies considering a worst-case scenario: assuming that all other agents act adversarially.

## A.2 ENVIRONMENTS

### A.2.1 STARCREFT II MICROMANAGEMENT BENCHMARK (SMAC)

SMAC (Samvelyan et al., 2019) is a combat scenario of StarCraft II unit micromanagement tasks. We consider a partial observation setting, where an agent can only see a circular area around it with a radius equal to the sight range, which is set to 9. We train the ally units with reinforcement learning algorithms to beat enemy units controlled by the built-in AI. At the beginning of each episode, allies and enemies are generated at specific regions on the map. Every agent takes action from the discrete action space at each timestep, including the following actions: no-op, move [direction], attack [enemy id], and stop. Under the control of these actions, agents can move and attack in continuous maps. MARL agents will get a global reward equal to the total damage done to enemy units at each timestep. Killing each enemy unit and winning the combat (killing all the enemies) will bring additional bonuses of 10 and 200, respectively. After training, the agents need to learn a feasible policy to win the battle even when there is an imbalance between the allies and the enemies. For example, in map 5m_vs_6m, agents need to learn to focus fire, i.e., jointly attack and kill enemy units one after another, to reduce the enemy force as soon as possible.

## A.2.2 PREDATOR-PREY (PP)

At the beginning of each episode, predators and prey spawn at random positions in a grid world. Agents can move in one of the four compass directions, remain immobile, or try to catch any adjacent prey. The prey can move and can be caught only if there are no empty grids around the prey and at least two predators adjacent to it execute the "catch" action concurrently. Once some agents successfully capture prey, the agents will receive a reward of 10, and both the agents and the prey will be removed from the grid. However, if an agent executes "catch" action and no prey is captured, all agents would receive a punishment of $-2$. These features further complicate the overall task that the agents are required to complete. In this paper, eight predators and eight prey are generated at random positions in a $10 \times 10$ grid world at the start of an episode. We consider all agents have a restricted field of vision with a range of 2 ($5 \times 5$ grids around the agent).

## A.3 MORE EXPERIMENTAL RESULTS

| Maps / Methods | | 2s3z | 3m | 3s_vs_5z | 5m_vs_6m |
|---|---|---|---|---|---|
| Attack Mode 1 | Vanilla QMIX | 97.7±0.3 | 97.9±0.2 | 89.5±2.0 | 68.5±1.8 |
| | Random QMIX | **99.2±0.8** | **98.3±0.7** | 98.3±0.7 | 73.1±5.4 |
| | One-agent QMIX | 84.2±2.8 | 92.9±1.7 | 96.3±0.9 | 65.0±4.1 |
| | ROMAO | 98.1±0.5 | 96.8±0.3 | **98.6±0.7** | **75.6±3.1** |
| Attack Mode 2 | Vanilla QMIX | 95.3±1.8 | 98.3±0.5 | 88.8±2.1 | 72.9±7.8 |
| | Random QMIX | **98.8±0.9** | **98.9±0.6** | 99.2±0.7 | 72.6±4.3 |
| | One-agent QMIX | 83.3±2.2 | 94.3±1.4 | 95.0±1.6 | 67.4±5.2 |
| | ROMAO | 97.9±9.8 | 95.3±2.1 | **99.3±1.3** | **77.1±2.1** |
| Attack Mode 3 | Vanilla QMIX | 95.4±1.3 | 0.0±0.0 | 39.3±3.0 | 21.2±2.2 |
| | Random QMIX | 98.1±0.8 | **98.9±0.9** | 77.5±2.9 | 63.1±3.4 |
| | One-agent QMIX | 75.6±2.5 | 96.9±0.7 | 91.8±1.5 | 55.6±3.5 |
| | ROMAO | **98.6±0.6** | 98.5±0.8 | **92.4±1.6** | **63.7±1.6** |

Table 2: Average test win rates of different methods under various attack modes with limited perturbation range of amount 2 in terms of the $\ell_1$ norm.

| Maps / Methods | | 2s3z | 3m | 3s_vs_5z | 5m_vs_6m |
|---|---|---|---|---|---|
| Attack Mode 1 | Vanilla QMIX | 89.1±1.1 | 76.7±1.3 | 21.1±3.8 | 20.8±1.8 |
| | Random QMIX | 96.9±0.6 | **96.5±1.5** | **96.7±0.7** | 12.2±0.8 |
| | One-agent QMIX | 80.8±2.4 | 92.3±2.3 | 28.3±3.6 | 14.4±2.4 |
| | ROMAO | **97.1±1.2** | 91.8±1.7 | 81.2±2.3 | **41.7±1.9** |
| Attack Mode 2 | Vanilla QMIX | 91.7±1.4 | 76.7±1.8 | 20.3±2.8 | 26.2±2.3 |
| | Random QMIX | **98.8±1.0** | **98.3±0.9** | **97.6±0.7** | **65.1±4.3** |
| | One-agent QMIX | 80.1±3.3 | 92.4±2.7 | 29.6±3.8 | 28.3±5.0 |
| | ROMAO | 96.8±1.3 | 93.3±1.7 | 80.6±2.2 | 51.1±3.4 |
| Attack Mode 3 | Vanilla QMIX | 1.4±1.0 | 0.3±0.2 | 48.0±3.4 | 0.0±0.0 |
| | Random QMIX | 12.2±1.2 | **91.2±2.5** | 44.0±2.0 | 0.0±0.0 |
| | One-agent QMIX | 32.5±4.1 | 88.7±2.6 | 83.5±4.0 | 0.0±0.0 |
| | ROMAO | **79.6±2.3** | 87.0±1.5 | **86.0±4.0** | **4.2±1.2** |

Table 3: Average test win rates of different methods under various attack modes with limited perturbation range of amount 10 in terms of the $\ell_1$ norm.

Table 2 shows the results of our experiments about the average test win rates and correspondent standard deviations of different methods under various attack modes with a limited perturbation range of amount 2 in terms of the $\ell_1$ norm. As the perturbation is relatively subtle, our ROMAO-

| Maps / Methods | | 2s3z | 3m | 3s_vs_5z | 5m_vs_6m |
|---|---|---|---|---|---|
| Attack Mode 1 | Vanilla QMIX | 73.6±2.4 | 57.7±0.5 | 0.1±0.2 | 0.8±0.8 |
| | Random QMIX | 86.8±2.6 | **95.3±0.8** | **90.0±1.5** | 9.7±2.7 |
| | One-agent QMIX | 71.0±2.3 | 62.2±3.2 | 0.1±0.3 | 0.8±0.8 |
| | ROMAO | **93.8±2.0** | 84.4±1.3 | 73.3±1.2 | **26.7±3.9** |
| Attack Mode 2 | Vanilla QMIX | 83.1±0.7 | 59.6±1.9 | 0.0±0.0 | 3.8±0.5 |
| | Random QMIX | **95.8±1.3** | **96.3±0.9** | **90.4±2.7** | **55.6±3.7** |
| | One-agent QMIX | 76.7±4.4 | 62.6±2.8 | 0.0±0.0 | 11.3±1.8 |
| | ROMAO | 95.6±2.6 | 83.3±1.7 | 76.4±3.0 | 29.7±1.8 |
| Attack Mode 3 | Vanilla QMIX | 0.5±0.2 | 0.0±0.0 | 40.5±3.2 | 0.0±0.0 |
| | Random QMIX | 0.4±0.3 | **90.4±1.9** | 24.0±2.9 | 0.0±0.0 |
| | One-agent QMIX | 0.0±0.3 | 47.2±3.0 | **76.4±3.6** | 0.0±0.0 |
| | ROMAO | **29.2±3.8** | 89.2±1.0 | 61.1±2.7 | **3.1±1.1** |

Table 4: Average test win rates of different methods under various attack modes with limited perturbation range of amount 20 in terms of the $\ell_1$ norm.

trained policy does not show a great advantage over other methods while still dominating in most scenarios.

Tables 3 and 4 show the results of our experiments about the average test win rates and correspondent standard deviations of different methods under various attack modes with a limited perturbation range of amount 10 and 20 in terms of the $\ell_1$ norm. From the experimental results, we can see that ROMAO can defend against different attacks. Although Random QMIX and One-agent QMIX perform better than ROMAO in some environments (e.g., 3s_vs_5z), they all experience a significant drop in performance in some other environments(e.g., 5m_vs_6m). On the contrary, the performance of ROMAO is relatively stable.

## A.4 SOFTWARE

We use the following software versions:

- Python 3.8
- SMAC 1.0 (Samvelyan et al., 2019)
- PyTorch 1.12.1 (Paszke et al., 2019)

## A.5 HARDWARE

We use the following hardware:

- NVIDIA RTX A4000
- 12th Gen Intel(R) Core(TM) i9-12900K

## A.6 IMPLEMENTATION AND HYPER-PARAMETERS

We implement ROMAO based on the author-provided implementation of HyAR (Li et al., 2021) and QMIX (Rashid et al., 2018). We list the hyper-parameters of the attacker's network, HyAR (Li et al., 2021) and ROMAO in table 5, the other parameters remain the same as the default for the pymarl framework (Samvelyan et al., 2019).

| | Hyper-parameters | Value |
|---|---|---|
| Attacker's Network | Actor learning rate | $3 \times 10^{-4}$ |
| | Critic learning rate | $3 \times 10^{-4}$ |
| | Batch size | 256 |
| | Optimizer | Adam |
| | Q-network | 3 layers ReLU activated MLPs with 256 units |
| | Policy Network | 3 layers ReLU activated MLPs with 256 units |
| HyAR | VAE training steps | $10^4$ |
| | VAE batch size | 128 |
| | HyAR batch size | 256 |
| | Discrete action dim | 8 |
| | Parameter action dim | 64 |
| ROMAO | Number of iterations | $3 \times 10^7$ |
| | Test interval | $10^5$ |
| | Test episodes | 20 |
| | Perturbation C | 10 |
| | Buffer size | 5000 |
| | Target update interval | 200 |

Table 5: The hyper-parameter setting of ROMAO.

