# OpenReview forum: "Robust Multi-Agent Reinforcement Learning against Adversaries on Observation"
_ICLR.cc/2023/Conference — Submitted to ICLR 2023_

### Official Review · Reviewer_8Gkv · 2022-10-23

**Confidence:** 4
**Correctness:** 3
**Technical Novelty And Significance:** 2
**Empirical Novelty And Significance:** 2
**Recommendation:** 5

**Clarity, Quality, Novelty And Reproducibility:**

Clarity: this paper is clearly presented.

Quality: the quality of this paper is okay but limited.

Novelty: the novelty of this paper is relatively limited.

Reproducibility: code and instructions are provided, although I did not run it.

**Strength And Weaknesses:**

Strengths:

1. The problem of observation attack in MARL environments is relatively new and interesting.
2. The paper provides attack and defense results in various scenarios, and the visualizations are good for understanding.
3. The paper is in general well-writte and easy-to-follow.
4. The idea of the defense module is interesting.

Weaknesses:

1. My major concern is on the novelty of the paper.
- The MARL attack problem is a straightforward extension of the single-agent attack problem, as the attack just need to learn an agent id in addition to the observation perturbation. Such an additional action output is easy to achieve with DNN parametrization. It would be more interesting and less trivial if the attacker attacks more than one agents simultaneously (e.g. perturb the observations of up to K agents).
- The alternate training idea is commonly used by many prior works [1,2,3] to improve the agent robustness, which makes the novelty of the paper limited. What is more, prior works focus on the robustness of a single victim agent, whereas this paper aims to improve the robustness of an MARL system. In this case, I would be worried about the efficiency of alternate training, given that training multi-agent policies is usually more difficult and less stable than training a single agent.

2. The experiment section looks interesting, but not adequate to justify the significance of the work. For example, what is the clean performance of the proposed training method (i.e., the reward of the ROMAO models when there is no attack)? This is an important metric for practical use of the robust training approaches [1,2]. It is possible that the model overly adapt to attacks while not achieving good natural performance.

3. The defense module is interesting, but the details are not illustrated much in the paper. This method is reasonable when the observations have redundant information. For examples, agents look at the same object from different angles. However, if the observation information of each agent does not overlap, such a reconstruction may not be achievable. In addition, since the agent does not know who is attacked, it is possible that the other agents' observations are corrupted. How do they deal with this case? More details of this defense would be appreciated.




[1] Huan Zhang, Hongge Chen, Duane S. Boning, and Cho-Jui Hsieh. Robust reinforcement learning on state observations with learned optimal adversary. ICLR, 2021.
[2] Yanchao Sun, Ruijie Zheng, Yongyuan Liang, and Furong Huang. Who is the strongest enemy? towards optimal and efficient evasion attacks in deep RL. ICLR 2022.
[3] Lerrel Pinto, James Davidson, Rahul Sukthankar, and Abhinav Gupta. Robust adversarial reinforcement learning. ICML 2017.

**Summary Of The Paper:**

This paper studies the adversarial robustness of MARL, with a focus on observation attacks. The proposed attacker can decide both which agent to attack and how to perturb its observation, via a hybrid-action attacker. Then the paper proposes a robust training approach by alternately training the attacker and the agents. An additional defense module is proposed which reconstructs the attacked observation based on the observations of teammates. Experiments in StarCraftII and Predator-Prey have shown the effectiveness of the proposed attacks and defenses.

**Summary Of The Review:**

This paper studies an interesting and important problem, but the proposed method is of limited novelty and may suffer from efficiency issues. The presentation of the paper is good, while the completeness of experiments can be further improved.


---- After rebuttal ---

Thank the authors for the responses. I think this line of research is indeed interesting. But I think the current work has relatively limited contribution, so I will maintain my score.

---

> ### Author Response · Authors · 2022-11-08
> **Response to Reviewer 8Gkv**
>
> Thanks for your thoughtful comments! We offer some clarification to your questions here and different parts we would try to improve in our future work.
>
> **Q1:** The novelty of this paper is limited and the situations under which two or more agents are attacked should be considered.
>
> **A1:** That is indeed an interesting line of research and we would focus on situations in which two or more agents are under attack in our future work. In addition, we will further improve our defense module since currently it only supports the reconstruction of a single agent's observation.
>
> **Q2:** About the metrics used to demonstrate the method's performance.
>
> **A2:** We use win rates as our metric of policy performance as this is commonly used when it comes to SMAC benchmarks. We did not show many results where there is no attack but we can see from Fig. 5(b) that ROMAO does not lose much performance when the perturbation range is 0. Illustrated in Tab. 2 in the appendix, ROMAO still keeps a good performance when the perturbation is small. We would include more results under no attack in our main text in our next revision to show that ROMAO can still achieve good natural performance.
>
> **Q3:** More details about the defense module need to be introduced.
>
> **A3:** The defense module needs communications between agents. As agents' observations can overlap, it is possible to reconstruct one agent's observation from its neighbors' observations. We train such a reconstructor during the centralized training phase. We will include more details about this in our next version.

---

### Official Review · Reviewer_gLvj · 2022-10-24

**Confidence:** 5
**Correctness:** 3
**Technical Novelty And Significance:** 1
**Empirical Novelty And Significance:** 2
**Recommendation:** 3

**Clarity, Quality, Novelty And Reproducibility:**

The problem studied is almost identical to the single-agent setting since (1) a fully cooperative MARL environment is considered; (2) the attacker is external to the system and can only target a single agent; (3) the attacker has full knowledge about the MDP model and the joint MARL policy. With these simplifications, the attacker's problem (when the MARL policy is fixed) is again single-agent MDP. The only difference compared with previous work on single-agent state perturbation is that now the attacker also needs to decide which agent to attack in addition to the perturbation action. The hybrid action space does not introduce significant new challenges, and the paper simply adopts the HyRL approach of Li et al. (2021) to address that.

The paper mentions very briefly a defense component where an agent reconstructs its observation from the observations of its neighboring teammates, which sounds interesting. However, no details are provided.

It looks like the paper only considers random attacks in the experiments according to the three attack modes defined in Section 5.2, which is inconsistent with the three attack types mentioned in Section 3. In particular, it is unclear why the paper does not consider the worst-case attack that responds optimally to the fixed MARL policy.

**Strength And Weaknesses:**

Recent work has focused on state perturbation attacks against a single RL agent. It is interesting to study the multi-agent setting. The alternate training approach for robust RL was proposed by Pinto et al. (2017) and has recently been applied to train a robust policy for single-agent state perturbation by Zhang et al. (2021). Due to its generality, the application of this approach to the multi-agent setting considered in the paper is straightforward. That being said, I am still surprised that the paper only provides a short description of the approach without providing any details.

There are two problems with the alternate training approach, none of which has been addressed in the paper. First, it has a very high training complexity due to the alternate training of the MARL policy and the attack policy, and it is unclear if the approach will converge. Second, the policy thus trained can be conservative. As shown in the experiments, the robust policy obtained this way is often weaker than the policy trained using adversarial training against random attacks.



**Summary Of The Paper:**

This paper studies observation perturbation attacks in cooperative multi-agent reinforcement learning (MARL), where the attacker can choose one agent to attack and inject noise with a bounded norm into its observations to minimize the long-term return of the system. The paper considers a test stage attack where a MARL policy has already been trained, and the attacker can observe both the underlying model and the MARL policy. The paper adopts an alternate training approach to train a robust defense policy and evaluates the approach in two MARL environments.

**Summary Of The Review:**

The paper considers a state perturbation attack that targets a single agent in a cooperative multi-agent reinforcement learning (MARL) system and adopts alternative training to obtain a robust policy. Due to the simplification of the threat model considered, the problem is similar to state perturbation against single RL. The proposed attack and defense methods thus follow known techniques with little adaptation and do not provide new insights into MARL security.

---

> ### Author Response · Authors · 2022-11-08
> **Response to Reviewer gLvj**
>
> Thanks for your inspiring comments! We offer some clarification to your questions here and hope they could be helpful.
>
> **Q1:** More detailed description of the alternating training approach should be added.
>
> **A1:** It is truly a critical part that we should emphasize in our main text. We will introduce the alternating training approach in more detail in our next version.
>
> **Q2:** What about the training complexity of this approach?
>
> **A2:** As the attacker adapts hybrid action space and a single-agent policy to perturb the observations of the multi-agent system, the training complexity for the attacker is not a burden. In addition, the multi-agent policy converges fast after a switch of training and we would include a more detailed training curve in our next revision.
>
> **Q3:** Why the results of worst-case attacks are not included?
>
> **A3:** Since we consider the situation of black-box attacks, we want to examine the performance of our robust policy under some general attacks like random perturbations. However, worst-case attacks are also critical to show the lower bound of our policies' performance and we would include that in our next revision.
>
> **Q4:** More details about the defense module need to be introduced.
>
> **A4:** The defense module needs communications between agents. As agents' observations can overlap, it is possible to reconstruct one agent's observation from its neighbors' observations. We train such a reconstructor during the centralized training phase. We will put more effort into this module in our future work and include more details about this in our next version.

---

### Official Review · Reviewer_G76o · 2022-10-31

**Confidence:** 4
**Correctness:** 3
**Technical Novelty And Significance:** 3
**Empirical Novelty And Significance:** 2
**Recommendation:** 3

**Clarity, Quality, Novelty And Reproducibility:**

The clarity of the paper is decent but there are still key details missing. There are not many adversarial training framework for MARL and this paper aims to advance this research direction which is good. T

**Strength And Weaknesses:**

Strengths:
- The new method employs a hybrid action space to select agent to attack along with the perturbation. The authors also incorporate encoder-decoder structure so that the learning is conducted on a latent repsentation space.
- The approach is shown to make the MARL agent robust to various attacks in SMAC and PP environments.

Weaknesses:
- My major concern is that the approach presented here is somewhat heuristic. For example, there is no discussion on the convergence guarantee when performing adversarial training. The authors presents an alternating training approach between the attacker and victim agents. I wonder if they happen to select this or the authors have tried other approach and this seems to work well. More details about this are welcome.
- Another concern is about how the attacker is trained. The authors provide 3 attack mode but do not clearly describe how attackers are trained under those modes. More details should be provided.
- The defense module is vaguely described and it is not easy to understand the detailed steps without going through the code. It would be great to have more details of this procedure.
- In a multi-agent setting, there is possibility that two or more agents get attacked simultaneously which is not considered in the paper. I wonder what the authors think about extending the current approach to this setting.

**Summary Of The Paper:**

The paper proposes a new training framework for multi-agent reinforcement learning (MARL) to improve the robustness of MARL agents by generating state/observation perturbation on strategically selected agent during training. They additionally propose a defense module to  prevent the MARL agent from being attacked by observation perturbation. Experiments on SMAC and predator-prey (PP) environmenst are presented to illustrate the effectiveness of the adversarial training framework.

**Summary Of The Review:**

I think the current paper is more like a technical report on what the authors did to perform adversarial training on a MARL setting. I do not see a systematic study into why the authors select such framework or any discussion on the convergence guarantee of the adversarial training framework. Some details are missing such as the training of attacker and the defense system.

---

> ### Author Response · Authors · 2022-11-08
> **Response to Reviewer G76o**
>
> Thanks for your inspiring comments! We offer some clarification to your questions here and hope they could be helpful. We also include some particular parts that we would try to improve in our future work.
>
> **Q1:** The approach is somewhat heuristic.
>
> **A1:** Adversarial training has been used in many fields, like image classification and reinforcement learning, and its effectiveness in improving agents’ robustness has been empirically shown in [1, 2]. The alternating training approach is chosen in consideration of the fact that we need to incessantly find the vulnerability of our multi-agent cooperative policies, as one round of training cannot ensure that our robust policy can deal with all different attacks. Concerning the convergence guarantee, we will put more effort into this part in our future work.
>
> **Q2:** How the attackers are trained listed in the table?
>
> **A2:** To show the robustness of our method against general attacks, these attacks tested are not trained and are randomly generated. Attack mode 1 denotes the attack in which the attacker randomly chooses an agent at every timestep and generates random perturbations. Attack mode 2 denotes the attack that at the start of an episode, the attacker randomly chooses an agent and generates random perturbations on that agent’s observation throughout the entire episode. Attack mode 3 denotes the attack in which the attacker randomly chooses an agent at the start of an episode and generates perturbations on only one random dimension of the agent’s observation.
>
> **Q3:** More details about the defense module need to be introduced.
>
> **A3:** The defense module needs communications between agents. As agents’ observations can overlap, it is possible to reconstruct one agent’s observation from its neighbors’ observations. We train such a reconstructor during the centralized training phase. We will include more details about this in our next version.
>
> **Q4:** What about situations in which two or more agents are attacked?
>
> **A4:** That is an interesting research direction and we plan to focus on this part in our future research to enrich our work.
>
> [1] Huan Zhang, Hongge Chen, Duane S. Boning, and Cho-Jui Hsieh. Robust reinforcement learning on state observations with learned optimal adversary. In *ICLR*, 2021.
>
> [2] Lerrel Pinto, James Davidson, Rahul Sukthankar, and Abhinav Gupta. Robust adversarial reinforcement learning. In *ICML*, pp. 2817–2826, 2017.

---

### Public Comment · ~Ezgi_Korkmaz2 · 2022-11-19
**Drawbacks and Shortcomings of Adversarial Training**

The below papers discuss the drawbacks and the shortcomings of adversarial training in deep reinforcement learning. Thus, it would be highly relevant to mention these studies below when referring to the adversarial deep reinforcement learning field; in particular, to adversarial training in deep reinforcement learning.

[1] Deep Reinforcement Learning Policies Learn Shared Adversarial Features Across MDPs. AAAI Conference on Artificial Intelligence, 2022.

[2]  Investigating Vulnerabilities of Deep Neural Policies. Conference on Uncertainty in Artificial Intelligence (UAI), 2021.

---

### Decision · Program_Chairs · 2023-01-20

**Decision:**

Reject

**Justification For Why Not Higher Score:**

Major concerns are a lack of motivation for the algorithm and unclear description of the framework.

**Justification For Why Not Lower Score:**

N/A

**Metareview: Summary, Strengths And Weaknesses:**

The paper presents a new Multi Agent RL (MARL) method to improve robustness by generating adversarial perturbations on the observation space during training. The reviewers find some of the ideas in hybrid action space and perturbation quite interesting. However, there are several concerns, from motivation (theoretical or descriptive), lack of training details, and an unclear presentation of the roles of different agents. Detailed comments are present in the reviewer's feedback. At this point, the reviewers do not find this work ready for publication at ICLR.